# De Novo Variant in the *KCNJ9* Gene as a Possible Cause of Neonatal Seizures

**DOI:** 10.3390/genes14020366

**Published:** 2023-01-31

**Authors:** Taisiya O. Kochetkova, Dmitry N. Maslennikov, Ekaterina R. Tolmacheva, Jekaterina Shubina, Anna S. Bolshakova, Dzhenneta I. Suvorova, Anna V. Degtyareva, Irina V. Orlovskaya, Maria V. Kuznetsova, Anastasia A. Rachkova, Gennady T. Sukhikh, Denis V. Rebrikov, Dmitriy Yu. Trofimov

**Affiliations:** Kulakov National Medical Research Center for Obstetrics, Gynecology and Perinatology, 117198 Moscow, Russia

**Keywords:** neonatal seizure, whole exome sequencing (WES), trio, KCNJ9, GIRK

## Abstract

Background: The reduction in next-generation sequencing (NGS) costs allows for using this method for newborn screening for monogenic diseases (MDs). In this report, we describe a clinical case of a newborn participating in the EXAMEN project (ClinicalTrials.gov Identifier: NCT05325749). Methods: The child presented with convulsive syndrome on the third day of life. Generalized convulsive seizures were accompanied by electroencephalographic patterns corresponding to epileptiform activity. Proband WES expanded to trio sequencing was performed. Results: A differential diagnosis was made between symptomatic (dysmetabolic, structural, infectious) neonatal seizures and benign neonatal seizures. There were no data in favor of the dysmetabolic, structural, or infectious nature of seizures. Molecular karyotyping and whole exome sequencing were not informative. Trio WES revealed a de novo variant in the *KCNJ9* gene (1:160087612T > C, p.Phe326Ser, NM_004983), for which, according to the OMIM database, no association with the disease has been described to date. Three-dimensional modeling was used to predict the structure of the KCNJ9 protein using the known structure of its homologs. According to the predictions, Phe326Ser change possibly disrupts the hydrophobic contacts with the valine side chain. Destabilization of the neighboring structures may undermine the formation of GIRK2/GIRK3 tetramers necessary for their proper functioning. Conclusions: We believe that the identified variant may be the cause of the disease in this patient but further studies, including the search for other patients with the *KCNJ9* variants, are needed.

## 1. Introduction

Seizures are a clinical manifestation of an abnormal, hypersynchronous discharge of a population of cortical neurons [1], and the risk of their development is highest in the neonatal period [2]. This is probably because at birth the brain is in a state of ongoing development, the balance of excitation and inhibition in it is still in the process of formation, and, therefore, can be more easily disturbed by various provoking factors [3].

The frequency of neonatal seizures (NS) is approximately 1–5.5 cases per 1000 live full-term newborns. The etiology of NS is quite diverse and includes hypoxic-ischemic encephalopathies, stroke, intracranial bleeding, metabolic or electrolyte disorders, infections, congenital malformations of the central nervous system, congenital metabolic disorders, and epilepsy syndromes, as well as cases of unclear etiology (up to 12%) [2,3,4].

Timely determination of the cause of neonatal seizures is important for choosing the proper treatment [5]. For this, the following methods are used: the analysis of clinical history, routine biochemical studies, neuroimaging (neurosonography (NSG)), magnetic resonance imaging (MRI), neurophysiological examination (electroencephalogram (EEG)), and other specialized tests [6], including genetic tests.

For genetic testing, both cytogenetic and molecular cytogenetic methods, as well as molecular genetic methods, can be used. Cytogenetic studies (karyotyping) make it possible to assess the presence of aneuploidies, unbalanced translocations, and ring chromosomes (for example, the ring chromosome 20 syndrome). However, they are limited by the size of the detected rearrangement (5–10 Mb). At the same time, chromosomal microarray analysis makes it possible to assess the presence of smaller deletions and duplications (100–300 Kb), and in the case of SNP arrays, to register a loss in heterozygosity. New generation sequencing technologies allow the use of both individual selected panels of genes (with increased test sensitivity instead of reducing the number of tested genes), and whole exome sequencing (with complete coding regions but at a risk of missing a variant in a poorly covered gene). Whole genome sequencing, despite the reduction in cost, does not have an extremely wide clinical application [7].

The choice of a specific approach depends on the confidence of the geneticist in the involvement of a particular gene in the observed phenotype. With high confidence in the correct interpretation of the phenotype, you can stop at the panel of genes, otherwise, whole exome sequencing is the method of choice [8].

Among patients with different types of epilepsy, the effectiveness of genetic testing varies with age of onset and severity of the disease [7].

The latter may be important both for the choice of treatment tactics and for the planning of subsequent pregnancies in a family: for instance, recessive inheritance is more typical for metabolic disorders, maternal inheritance is typical for mitochondrial and X-linked diseases [5], accounting for the increased risks in the following pregnancies that may be reduced using preconception, prenatal, or rapid postnatal diagnostics in a family. In turn, channelopathies tend to be dominantly inherited, and severe forms are more likely to be the result of a de novo mutation, showing low risk for following pregnancies. However, disturbances of ion channels described for potassium, sodium, and calcium channels are responsible for most forms of monogenic epileptic syndromes, from benign neonatal seizures to forms of epilepsy not limited to childhood [9]; therefore, the establishment of an accurate genetic diagnosis affects the long-term prognosis and treatment planning for the patient.

According to the published data, knowing the genetic diagnosis may affect the treatment for almost 40% of diagnosed patients. The diagnostic yield of whole exome sequencing for pediatric patients with seizures is estimated to be about 30% [10], varying considerably between studies [11]. It is important to note that while less than a half of human genes are currently linked to human disorders, new gene–diseases associations are reported every year, allowing more patients to receive a diagnosis. While one-step analysis does not always yield a meaningful result, it is important to consider other options, such as expanding WES to the analysis of trio sequencing, genome sequencing, and copy number variation analysis, exploring the segregation in families with multiple affected siblings.

While the cost of sequencing has been rapidly reducing since the beginning of its use for genetic diagnostics, it is now becoming a first-line choice for many patients. Moreover, there are some projects where WES is explored as a screening method for apparently healthy newborns [12].

Here, we report a neonate with neonatal seizures, who carries a de novo variant in the *KCNJ9* gene (1:160087612T > C, p.Phe326Ser, NM_004983), for which, according to the OMIM database, no association with the disease has been described to date, discovered during the whole-exome newborn screening project “EXAMEN” (ClinicalTrials.gov Identifier: NCT05325749, performed for all newborns at the Kulakov National Medical Research Center for Obstetrics, Gynecology and Perinatology, Moscow, Russia).

## 2. Materials and Methods

This case study was carried out according to the Code of Ethics of the World Medical Association (Declaration of Helsinki). Participants (parents) gave written informed consent for the use of any data for scientific purposes. The whole exome newborn screening project “EXAMEN” study protocol was reviewed and approved by the Ethics Committee of the Kulakov National Medical Research Center for Obstetrics, Gynecology and Perinatology (Protocol No.9 from 22 October 2020).

Within the framework of the “EXAMEN”, the cord blood was collected from all newborns whose parents signed an informed consent form to participate in the project. For newborns without phenotypic features, a screening was carried out for the presence of monogenic diseases. For newborns with developmental features, suspicious of the presence of hereditary pathology, the cause of the disease was carried out, and additional studies were prescribed if necessary.

In August 2021, at the Kulakov National Medical Research Center for Obstetrics, Gynecology and Perinatology (Moscow, Russia), a full-term girl (weight 3306 g, body length 53 cm, 8–9 points on the Apgar scale) was born to a 29 year old woman (pregnancy without complications, first timely physiological delivery).

On the 3rd day of life, the newborn experienced increased excitability, anxiety during palpation of the head and at rest, intense crying, and repeated paroxysms of focal tonic convulsions with a predominance of limb hypertonicity. The EEG presented epileptiform activity in the form of separate high-amplitude positive sharp waves and acute-slow wave complexes, with index and amplitude predominance in the frontal leads of both hemispheres. Considering the examination data, the newborn was diagnosed with convulsive syndrome. Diazepam was administered for anticonvulsant purposes with a short-term positive effect, and the infant was transferred to the neonatal intensive care unit (NICU).

On the 4th day of life, the infant developed a series of generalized convulsive seizures, accompanied by decompensated cardio-respiratory status. Combined anticonvulsant therapy with valproic acid and benzodiazepine was prescribed, and the infant was put on artificial lung ventilation (ALV). During the observation period in the NICU, the newborn experienced repeated episodes of convulsive seizures that required an increase in the dosage of valproic acid. The infant was diagnosed with neonatal seizures of unknown etiology, and differential diagnosis was made between symptomatic neonatal seizures (structural, metabolic, infectious) and benign neonatal seizures.

According to the results of brain MRI, there was no structural pathology. Study of cerebrospinal fluid also revealed no abnormalities. The convulsive seizures stopped under the therapy, and according to EEG, no patterns of epileptic seizures were recorded in dynamics. The respiratory support was canceled on the 8th day of life, benzodiazepine on the 11th day of life, and on the 13th day the infant was transferred to the neonatal pathology department.

On the 16th day, the infant’s condition worsened, presenting with CNS depression, significantly decreased motor activity, muscle hypotonia, hyporeflexia, and, therefore, the child was transferred to the NICU. Upon admission, she presented with a depression of consciousness to coma of degrees 1–2 and the absence of spontaneous breathing that required respiratory support up to 22 days of life. The concentration of valproic acid in the blood serum was increased (552.65 μg/mg against normal 50–100 μg/mg); therefore, the drug was changed to phenobarbital. When assessing the level of ammonia in the blood serum, its significant increase was noted (859.3 μmol/L against normal 33–95 μmol/L); therefore, the infant was put on 24 h protein-free diet, and sodium benzoate was prescribed to stop the hyperammonemia crisis. Under this therapy, the ammonia level returned to normal. On the 19th day of life, the result of tandem mass spectrometry (TMS) for the spectrum of amino acids and acylcarnitines in the blood was obtained; it revealed increased levels of C12 and C12:1 and C14:1, as well as in the ratio C14:1/C2 and C14:1/C16, but the results of the repeated testing were within normal values. A diagnosis of transient hyperammonemia was made with genesis most likely associated with the use of valproic acid. EEG presented a low index, epileptiform activity in the form of acute-slow wave complexes and individual sharp waves. Ictal epileptic activity and EEG patterns of a convulsive syndrome were not registered.

On the 25th day of life in a state of moderate severity due to CNS depression, the infant was transferred to the neonatal pathology department for further treatment. The infant presented with positive dynamics. Repeated brain MRI revealed no abnormalities. According to the EEG series, epileptiform activity was not recorded. In a satisfactory condition on the 41st day of life, the infant was discharged for further outpatient follow-up by a neurologist.

At the age of 2 months, the infant had episodes of paroxysmal conditions of non-epileptic origin; according to EEG data, epileptiform activity was not registered. The child’s dynamics were investigated by a neurologist at the ages of 2, 5, 6, 7, and 10 months. In the 9th month of life, anticonvulsant therapy with phenobarbital was canceled. At the age of 10 months, no recurrent episodes of paroxysmal conditions were reported; according to EEG sleep monitoring, typical epileptiform activity was not registered.

The child was examined by a geneticist who recommended a search for mutations in the genes associated with neonatal seizures. Whole exome sequencing (WES) of the proband revealed no relevant variants, so sequencing was extended to the trio including the child’s parents.

The absence of clinically significant DNA copy number variation (CNV) was confirmed by chromosomal microarray analysis (CMA; ThermoFisher CytoScan750K array, Waltham, MA USA). The WES was performed using a NovaSeq 6000 instrument with Illumina^®^ (San Diego, CA, USA) DNA Prep (S) Tagmentation, IDT^®^ (San Diego, CA, USA) for Illumina^®^ DNA/RNA UD Indexes, xGen Exome Research Panel v2 reagent kits the reagent kits being used according to the producer’s instructions. The whole exomes were sequenced with coverage 87.99x, 208.8x, and 220.2x (the described variant was covered 188x, 531x, and 468x in the proband, mother, and father, respectively). In the proband sample variant, the read to total read ratio was 94/188. The variant was verified by Sanger sequencing. Bioinformatics analysis of the exome data was carried out using proprietary software, including sequence alignment to the reference genome GRCh38 (hg38), variant calling, and quality filtering. Annotation using the Ensembl Variant Effect Predictor (VEP) and a number of variant significance prediction algorithms (SIFT, PolyPhen-2, SpliceAI, CADD, REVEL) was carried out for all (filtered by quality) variants. [13,14,15,16,17]. The Genome Aggregation Database (gnomAD, (https://gnomad.broadinstitute.org/, accessed on 12 October 2022) was used to estimate the population frequencies of the identified variants [18]. To assess the clinical relevance of the identified variants, the OMIM, ClinVar, LOVD and other specialized databases (if available), as well as data from the literature, were used. ACMG criteria were used for data analysis.

The GRCh38 version of the reference genome was used. The de novo status of the variant found in the proband was confirmed by the whole exome trio method (using the parental exomes). The relationship between the proband and parents was confirmed using the plink 1.9 (https://www.cog-genomics.org/plink/) and VCFtools 0.1.16 (https://vcftools.github.io/).

## 3. Results

The analysis of the whole exome trio revealed a heterozygous missense variant in the *KCNJ9* gene (1:160087612T > C, GRCh38, p.Phe326Ser, NM_004983) that arose de novo (Figure 1). The figure shows that only the proband has the variant, while it was read by both forward and reverse reads, and the proportion of reads with an alternative allele is close to 50%, which indicates the reliability of the data obtained using exome sequencing. Pathogenicity predictors classified the variant as likely pathogenic (PolyPhen-2: 0.653, Sift: 0.0, CADD: 26.3, REVEL: 0.632). The variant was not registered in the control sample of gnomAD (https://gnomad.broadinstitute.org/, accessed on 12 October 2022), and, according to the same control sample, the *KCNJ9* gene is depleted in missense variants (Z = 4.22, o/e = 0.26), which suggests probable pathogenicity of variants of this type.

To date, the *KCNJ9* gene has not been associated with monogenic disorders, which explained the need for trio sequencing: to assess the significance of variants in genes with uncertain clinical significance, the origin of variants serves as an important criterion of pathogenicity (de novo status for dominant and trans position for recessive disorders).

To search for other patients carrying de novo variants in the *KCNJ9* gene, we used known databases such as GeneMatcher, denovo-db, Undiagnosed Network, Mastermind and Google search. So far, we have not received any information regarding similar phenotypes in patients with *KCNJ9* variants, or any data against the assumption of the pathogenicity of the observed variant.

The *KCNJ9* gene is located on chromosome 1 and consists of 9026 base pairs (NM_004983) (https://www.ncbi.nlm.nih.gov/gene/3765, accessed on 12 October 2022). The gene contains three exons, one of which is non-coding.

The *KCNJ9* gene encodes a G-protein-coupled inwardly rectifying K^+^ channel sub-unit 3 (GIRK3, UniProt entry Q92806) of unknown structure [19]. According to UniProt, the Phe326 residue was not reported to have any specific function such as ligand binding or forming essential structural motifs. Therefore, to gain an insight into the mechanics of a possible disruption of the KCNJ9 protein by the observed variant, it was necessary to predict its spatial structure. Currently, there are few approaches used in protein modeling, so first we tried AlphaFold software that has been developed to predict the three-dimensional structure of the proteins [20,21]. This computational approach is able to predict the protein structure even when no homologous structure is known. Using AlphaFold, we were able to build a three-dimensional model of KCNJ9, however, some parts of the protein sequence were obviously not folded, so we turned to an “old but gold” method. Using protein sequence alignment, we searched for homologs to serve as a template for building a 3D structure. Close homologs of human GIRK3 with known structures identified with BLASTP [22] are presented in Table 1. We constructed the GIRK3 structure with HHpred/Modeller [23,24] based on hidden Markov models to predict the protein structure using pairwise comparison. For that we have chosen the 3sya crystal structure of GIRK2 (76% identity) as a template. After the structure was built, the atomic-level energy minimization was performed allowing for an accurate prediction of the wild-type protein structure. Figure 2 and Figure 3 demonstrate that Phe326 is in the cytoplasmic domain of GIRK3 and corresponds to tyrosine in GIRK2, IRK1, and IRK2 homologs. The phenylalanine and tyrosine residues have similar biochemical properties, they both have a large hydrophobic side chain. According to the multiz-100-way alignment [http://hgdownload.soe.ucsc.edu/goldenPath/hg19/multiz100way/] of a protein sequence in 100 vertebrates, the phenylalanine/tyrosine residue at position 326 is conserved among different species. The 326 residue forms hydrophobic contacts with the valine side chain (or threonine in GIRK2) and aliphatic carbons of the arginine side chain (Figure 2b), and the Phe326Ser mutation in GIRK3 modeled with the Swiss-PdbViewer software [25] presumably disrupts these interactions (Figure 2c).

**Table 1 genes-14-00366-t001:** Close homologs of human GIRK3 with known structures found with BLASTP.

Protein	Organism	Identity, %	Query Coverage, % ^1^	PDB ID ^2^
GIRK2	*Mus musculus*	76	84	3sya [26]
IRK2	*Gallus gallus*	57	84	3jyc [27]
IRK1	*Homo sapiens*	55	86	7zdz [28]

^1^ Percentage of the query sequence length that is included in the alignment. ^2^ Entry in Protein Data Bank.

**Figure 2 genes-14-00366-f002:**
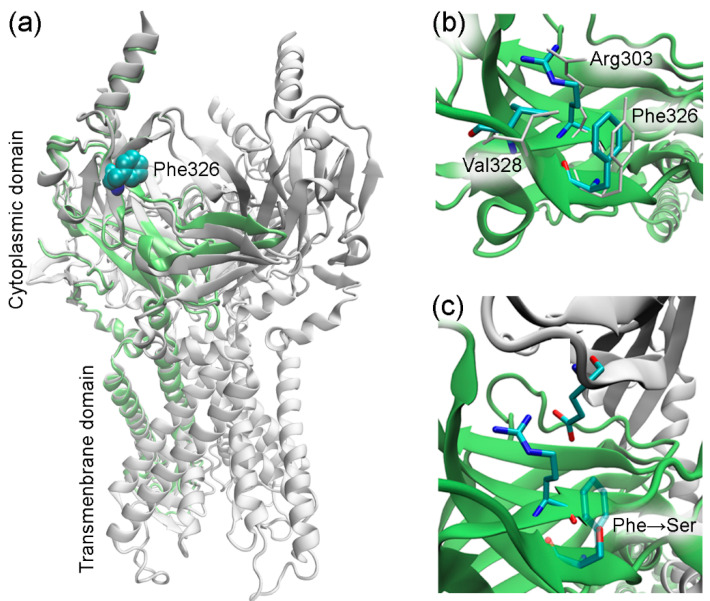
Homology modeling of GIRK3. (**a**) Modeled GIRK3 subunit. The template GIRK2 structure (3sya, tetrameric assembly) is shown in white, and GIRK3 is shown in lime. (**b**) Molecular environment of the residue 326 in GIRK3 (colored by atom type) and GIRK2 (white). (**c**) Phe326Ser substitution in GIRK3. The GIRK2 subunit (3sya) comprising the glutamate residue is shown in white. The figure was prepared using VMD [29].

**Figure 3 genes-14-00366-f003:**
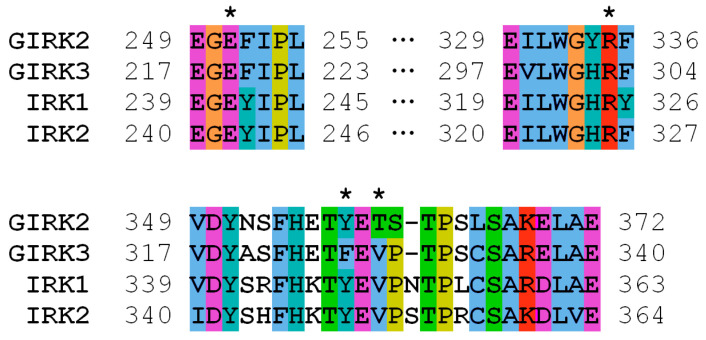
Multiple sequence alignment of human GIRK3 and its close homologs (UniProt entries P48051, Q92806, P63252, and Q14500). Residues 219, 303, 326, and 328 (GIRK3 numbering) are marked with asterisks.

## 4. Discussion

GIRK G-protein-coupled potassium channels of internal rectification, which have an inhibitory effect on neurons, dysregulate the activity of these channels and, thus, contribute to the development of various neurological diseases and disorders [30]. GIRK1-4 proteins are encoded by the KCNJ3 (GIRK1), KCNJ6 (GIRK2), KCNJ9 (GIRK3), and KCNJ5 (GIRK4) genes and are homologous subunits (also conserved across species, such as humans and mice, which is helpful for creation of mouse models), each of which has features of regulation and tissue and subcellular distribution [31], so that by composing various homo- or heterotetramers, these proteins can form channels with different properties and localization [32].

The distribution of GIRK4 expression in tissues differs from the other three proteins of the group (https://www.proteinatlas.org/, https://www.omim.org/entry/600734, accessed on 12 October 2022) [33,34]. It is known that GIRK1-3 are actively expressed in brain and form heterotetramers GIRK1–GIRK2, GIRK2–GIRK3, and GIRK1–GIRK3 and homotetramers GIRK2–GIRK2 [35]. For proteins of this group, the possibility of mutual regulation has been described: in GIRK2-/- knockout mice, GIRK1 expression in the brain is significantly reduced [36]. The regulatory role of GIRK3 is indicated by its presence in a structure unique for this group of proteins; a lysosomal target sequence (Tyr-Trp-Ser-Ile, or “yWSI”, residues 350–353), which activates the degradation of GIRK channels and, thus, inhibits their expression on the plasma membrane [37]. GIRK2, which forms channels together with GIRK3, has been associated with neurological disorders in model animals and in humans: spontaneous seizures have been described in GIRK2-/- and GIRK2.3-/- knockout mice [36,38], and in humans variants in the *KCNJ6* gene cause Keppen–Lubinsky syndrome, a severe hyperkinetic disorder (https://www.omim.org/entry/600877, accessed on 12 October 2022).

The amino acid residue Phe326 in GIRK3 that is altered in our proband is located close to the lysosomal target sequence 350–353. However, a C-terminal region comprising this sequence is likely to be disordered, as observed in available structures of GIRK3 homologs. For example, residues 344–393 (GIRK3 numbering) are missing in the 7zdz structure of IRK1. It is, therefore, unlikely that the amino acid residue at position 326 interacts with residues 350–353 and can regulate lysosomal targeting. On the other hand, disrupting hydrophobic interactions accompanied by the Phe326Ser mutation may hinder the formation of GIRK2/GIRK3 tetramers. The Phe326 side chain stabilizes the conformation of the neighbor residue Arg303 via favorable hydrophobic contacts, and the substitution of phenylalanine with serine apparently results in loss of the interaction with aliphatic carbons of Arg303 (Figure 2c). This might destabilize the polar interaction of the Arg303 guanidinium group with a glutamate residue (Glu251/219 in human GIRK2/GIRK3) of another protein subunit in the subunit assembly shown in Figure 2c. Since the above-mentioned arginine and glutamate residues are conserved in GIRK2 and GIRK3 (as shown in Figure 3), this polar interaction is allowed between both wild-type GIRK3–GIRK3 and GIRK3–GIRK2 subunits in the tetrameric assembly.

As far as we know, there are no reported data on disease association for Phe326 variants in GIRK3, as well as for corresponding positions in human homologs (Tyr358 in GIRK2, Tyr348 in IRK1, and Tyr349 in IRK2). We hypothesize that the Phe326Ser variant in the GIRK3 protein could lead to a disruption of its expression on the cell surface and subsequent neurological disorders.

## 5. Conclusions

This clinical case demonstrates a possible association of a single nucleotide substitution in the *KCNJ9* gene with neurological disorders (neonatal seizures or a syndrome that has neonatal seizures among its early manifestations). However, further research is needed to clarify the nature of this association and the disease caused by disorders in this gene.

Describing new gene–disease associations is important to expand the possibilities of genetic diagnostics in the future, and this case shows how WES trio analysis can be useful in finding such associations.

## Figures and Tables

**Figure 1 genes-14-00366-f001:**
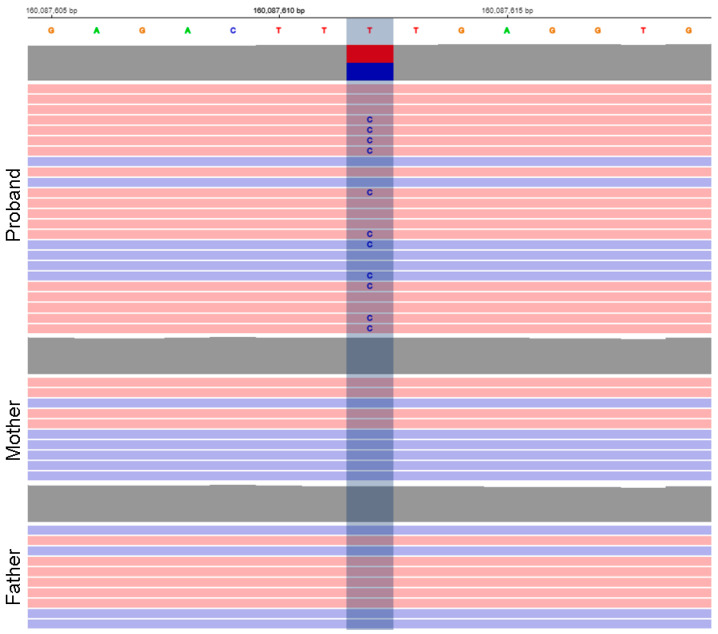
Variant view in the proband and parents in IGV viewer showing its de novo origin.

## Data Availability

No new data were created or analyzed in this study. Data sharing is not applicable to this article.

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
