# Peer review of "De Novo Variant in the *KCNJ9* Gene as a Possible Cause of Neonatal Seizures"

_genes, 2023, doi:10.3390/genes14020366_

Round 1

Reviewer 1 Report

This article nicely present case report of a girl with seizures. Trio study is a latest method to find denovo mutation. However, one thing that if you could do is Sanger sequencing of the variant. We always validate our result of exome sequencing with sanger sequencing.  

Author Response

Reviewer 1:

However, one thing that if you could do is Sanger sequencing of the variant. We always validate our result of exome sequencing with sanger sequencing.

Answer:

Indeed, the Sanger method is the gold standard for confirming genetic findings. Following the comments of the reviewers, over the past 10 days, we have carried out the determination of the mutation by the Sanger method and confirmed the NHS data. Corresponding changes have been made in the “Materials and Methods” section.

Reviewer 2 Report

Summary: The case report described a new family with a de novo missense variant in the KCNJ9 gene that contributes to the neonatal seizures phenotype. The report described detailed clinical phenotype, which is helpful. However, the molecular diagnosis needs to be elaborated. What is the variant read to total read ratio at the position? Does the segregation of the de novo variant verified with an orthogonal method like Sanger sequencing/ddPCR? The IGV view of the de novo at the position is helpful and necessary to be included. The variant is not observed in gnomad database.  What about the predicted consequence based on combined computational prediction algorithms, e.g, CADD or REVEL score? The amino acid change is close in sequence to the lysosomal target sequence; What about in the 3D structure?

Minor review:

1.      Which version of the reference genome does the variant refer to? (GRCh38?)

Author Response

Reviewer 2:

Q2-1: What is the variant read to total read ratio at the position?

Answer Q2-1:

The variant read to total read ratio at the position is: 94C/188.

Q2-2: Does the segregation of the de novo variant verified with an orthogonal method like Sanger sequencing/ddPCR?

Answer Q2-2:

Indeed, the Sanger method is the gold standard for confirming genetic findings. Following the comments of the reviewers, over the past 10 days, we have carried out the determination of the mutation by the Sanger method and confirmed the NHS data. Corresponding changes have been made in the “Materials and Methods” section.

Q2-3: The IGV view of the de novo at the position is helpful and necessary to be included.

Answer Q2-3:

We have added the IGV view of the de novo at the position to the text.

Q2-4: The variant is not observed in gnomad database. What about the predicted consequence based on combined computational prediction algorithms, e.g, CADD or REVEL score?

Answer Q2-4:

This is a very helpful remark. CADD: 26.3 was used and we added appropriate information in the text.

Q2-5: The amino acid change is close in sequence to the lysosomal target sequence; What about in the 3D structure?

Answer Q2-5:

We planned to make the 3D structure in the next publication, but managed to complete it according to the comments on this article. We added information about 3D structure modeling in the text.

Q2-6: Which version of the reference genome does the variant refer to? (GRCh38?)

Answer Q2-6:

Exactly GRCh38 was used.